# End to End Alignment Learning of Instructional Videos with Spatiotemporal Hybrid Encoding and Decoding Space Reduction

**Lin Wang** †, **Xingfu Wang** †, **Ammar Hawbani** *,† and **Yan Xiong** †

School of Computer Science and Technology, University of Science and Technology of China, Hefei 230027, China; xiaquhet@mail.ustc.edu.cn (L.W.); wangxfu@ustc.edu.cn (X.W.); yxiong@ustc.edu.cn (Y.X.)
* Correspondence: anmande@ustc.edu.cn; Tel.: +86-187-5691-8792

**Abstract:** We solve the problem of how to densely align actions in videos at frame level, with only the order of occurring actions available, in order to save the time-consuming efforts to accurately annotate the temporal boundaries of each action. We propose three task-specific innovations under this scenario: (1) To encode fine-grained spatiotemporal local features and long-range temporal patterns simultaneously, we test three popular backbones and compare their accuracy and training times: (i) a recurrent LSTM; (ii) a fully convolutional model; and (iii) the recently proposed Transformer model. (2) To address the absence of ground truth frame-by-frame labels during training, we apply connectionist temporal classification (CTC) on top of the temporal encoder to recursively collect all theoretically valid alignments, and further weight these alignments with frame-wise visual similarities, in order to avoid a significant number of degenerated paths and improve both recognition accuracy and computation efficiency. (3) To quantitatively assess the quality of the learned alignment, we apply a comprehensive set of frame-level, segment-level, and video-level evaluation measurements. Extensive evaluations verify the effectiveness of our proposal, with performance comparable to that of fully supervised approaches across four benchmarks of different difficulty and data scale.

**Keywords:** temporal video segmentation; temporal video alignment; connectionist temporal classification (CTC); transformer; convolutional neural networks (CNNs); computer vision

## 1. Introduction

Fine-grained temporal action segmentation/alignment [1–3] is important in many applications, such as daily activity understanding, human motion analysis, and surgical robotics, to name a few. Given a video of length $T$, $x = (x_1, \cdots , x_T)$, the goal of temporal segmentation/alignment is to localize each occurrence of a given action $a_n \in \mathcal{A}$ in the time domain. A frame-to-action segmentation/alignment can be mathematically defined densely, as a sequence of occurring action labels at every frame in a video $\pi = (\pi_1, \cdots , \pi_T)$, $\pi_t \in \mathcal{A}$; or sparsely, as a set of temporally segmented clips $\ell = (\ell_1, \cdots , \ell_N)$, $\ell_n \in \mathcal{A}$, with each segment associated with a start time, finish time, and label. $N = |\ell|$ is the length of the transcript sequence; note that usually $T \gg N$ since the sampling frequency of the machine at the encoding end is orders of magnitude higher than the granularity of the manual labeling at the decoding end.

The difference between the tasks of segmentation/alignment is that during training, only the ordered sequence of video-level actions $\ell$ (defined as transcript [4–6]) is available for alignment, while the classes of each frame are given for segmentation. Figure 1 shows the training/testing settings in the task of temporal action alignment learning, as compared to temporal action segmentation, where accurate dense labels of each frame are also available during training.

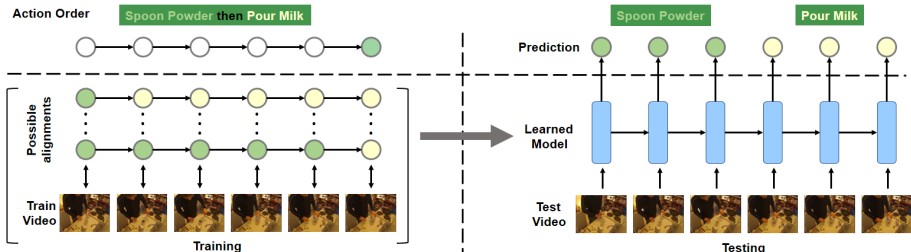

**Figure 1.** The training/testing settings in the problem of action labeling through temporal alignment learning. (**Left**) During training, only the order of the occurring actions is given, the model is learned by maximizing the probability of all possible frame-to-label alignments. (**Right**) During testing, no annotation is given, as the learned model already encodes the temporal structure of data, it predicts the actions frame by frame without any further information.

We also restrict our focus on modeling instructional videos with relatively stable background, usually composed of dozens of actions lasting minutes, recorded in a kitchen or surveillance setup. Under this setting, research focus can be saved from variances in extrinsic shooting conditions, and concentrates upon the major challenges of the task.

### 1.1. The Motivation of Long-Term Temporal Encoding

The first challenge lies in the relatively large temporal search space, resulting from long-range temporal dependencies and flexible temporal length, compared to action recognition (also known as action classification). In general, action alignment is more challenging than action recognition, for the following reasons:

- In action recognition, temporal localization is not taken into account, where input samples are previously truncated to contain exactly the temporal span of a certain target action, leading to relatively short inputs (e.g., 2~20 s in UCF101 datasets). In frame-wise action alignment, inputs generally last minutes or hours. In action recognition, background class is not taken into account, where input samples may not contain any of the target actions.
- In action alignment, dependencies can also last temporally across seconds or even minutes. The types of temporal dependencies include individual action durations, pairwise compositions between consecutive actions, and long-term compositions lasting across multiple sub-actions. As an example of a cooking instance, when cutting a potato, it is difficult to recognize what is being cut because it tends to be occluded by hands holding it. The recognition of frames where the potato is being cut shares dependencies with previous frames where the potato is being taken out before being cut.
- In action recognition, only one label needs to be assigned to the whole video, whereas the action alignment task needs to densely assign a label to each frame. Consider a video instance consisting of 20 frames sharing the same class.
- In action recognition, even if a convolution network correctly predicts only for 10 frames, it is still very likely to correctly predict the whole video. A powerful temporal encoder, illustrated in Figure 2, on top of a convolution network would not bring any improvement in this case, because per video labels do not change whether or not per frame labels are neighboring. In action alignment, 10 accurate but not neighboring predictions would lead to over-segmentation error with 20 segments. Bidirectional temporal encoders would be motivated to predict neighboring segments, because they encode late samples together with the early ones for final judgement, leading to fewer false positives compared to a purely convolutionnetwork (see Figure 3).

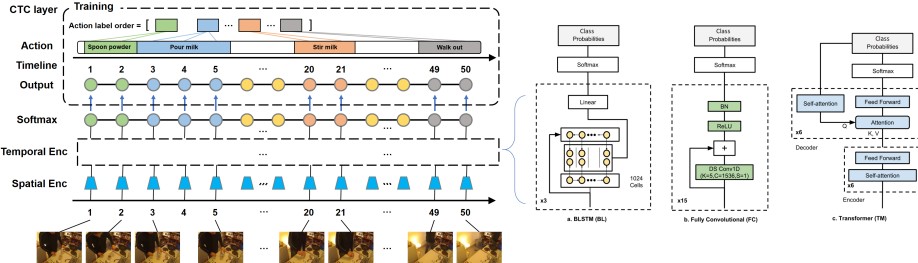

**Figure 2.** The proposed Hybrid-CTC structure consists of a hybrid encoder with a CTC output layer. See Table 3 for architectural details of different temporal encoders. See Figure 4 and Section 3.4 for how the CTC layer aggregates all possible input–output alignments. See Section 4.1 for how to sub-sample frames from raw video as pre-processing before submitting to CNN encoders.

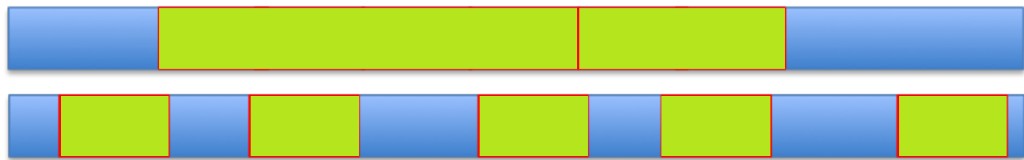

**Figure 3.** Similar frame-wise accuracy may have large qualitative differences. The top row generates neighbouring segments. The bottom row generates the same proportion of segments but they are non-contiguous, leading to numerous over-segmentation errors.

### 1.2. The Motivation of Hybrid Spatiotemporal Encoding

The second challenge lies in the simultaneous encoding of low-level fine-grained spatiotemporal features together with high-level long-range temporal patterns. Appearance information, which can be regarded as visual features presented by static composing frames without taking temporal order into account, also serves as vital to the start and finish boundary indicators of actions (e.g., 'cutting a tomato' is often only subtly different from 'peeling a cucumber' spatially, such as the food types 'tomatoes' or 'cucumbers' and food states 'sliced', 'diced', 'peeled', and so on).

We experiment with three deep neural network models for encoding. In each case the encoder consists of two modules (or sub-networks): a spatiotemporal visual module Table 2 encoding per frame into one feature vector, and a temporal module Table 3 encoding the sequence of per-frame feature vectors into a sequence of sub-action labels. The visual module is common across the three models; they only differ in the temporal module (Figure 2).

### 1.3. The Motivation of Decoding Space Reduction

The third challenge lies in that manual annotations with per-frame action labels for accurate training are too expensive to be feasibly applicable in practice. Automatic extractions from instructional transcriptions [4–6] can provide label sequences of occurring actions (hereafter referred to as 'transcript'), without the accurate start and end frame for each action, which further impose new challenges in two ways:

- The first challenge is that densely aligning thousands of frames to a few sub-actions results in very large search spaces of possible alignments.
- The second challenge is that there exist degenerated alignments that are visually inconsistent.

We introduce CTC [7] (Figure 2) to evaluate all alignments efficiently in one-pass recursive traverse, and further incorporate frame-level visual similarities to down-weight trivial paths which deteriorate performance. Figure 4 shows the difference of training strategies under different granularities of supervision.

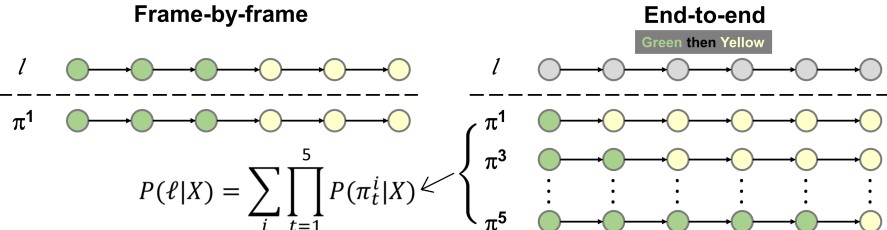

**Figure 4.** Blank nodes indicate unlabeled frames. CTC addresses the problem of unknown labels by aggregating associate paths (Equation (4)).

## 2. Related Work

We mainly focus on related deep end-to-end approaches, which can be grouped into two research directions according to different annotating granularity—that is, at frame-level (segmentation) and video-level (alignment) supervision:

### 2.1. Action Segmentation

Before deep neural solutions, the most prevalent were statistical models [8–10] based on conditional independent assumption between segments, which ignore long-range dependencies and have been generally outperformed by the state of the art (see Table 1 for more quantitative performance comparison). MSB-RNN [11] uses a two-stream network and bi-directional LSTMs to learn representations and capture dependencies between video chunks, respectively. ED-TCN [12] uses temporal convolutions and pooling layers within an encoder–decoder architecture to learn long-range dependencies between frames. TDRN (temporal deformable residual network) [13] proposes two parallel temporal streams, facilitating temporal segmentation at local, fine-scale, and multiple long-range scales, respectively, for improving the accuracy of frame classification. TCED [14] introduces a learnable bilinear pooling in the intermediate layers of a temporal convolutional encoder–decoder net, in order to capture more complex local statistics than conventional pooling. ASRF [15] proposes to alleviate over-segmentation errors by detecting and refining action boundaries with a dedicated boundary regression module and a wider temporal receptive field.

One problem in this context arises from the fact that the annotation to mark action boundaries for training is very time- and cost-intensive, leading to recent efforts trying to train classifiers without exact start and end frames of the related action classes (which is our case); the goal of this work is to infer frame boundaries given only an ordered list of the occurring actions.

### 2.2. Action Alignment

Compared with full supervision, there are only a few approaches that rely solely on video-level class labels to localize actions in the temporal domain. GRU + HMM [1] proposes a combination of a recurrent neural network and a probabilistic model to inference over long sequences for a temporal alignment. TCFPN + ISBA [2] proposes a pyramid temporal convolutional network, iteratively updated by a training strategy named ISBA (iterative soft boundary assignment), to align action sequences with frame-wise action labels in a more efficient and scalable fashion. CDFL [3] proposes a constrained discriminative forward loss upon GRU + HMM ground. Duration network [16] treats the the remaining duration of a given action as a predictable distribution conditioned on the type of action, and obtains the best alignment by maximizing its posterior probability.

In general, these methods are still more or less hand-engineered, involving statistical components designed on prior knowledge, and rely on sophisticated techniques to improve performance. To our best knowledge, the inherent integration of visual similarity into the CTC output layer with encoder–decoder spatiotemporal transducers sets our method apart

from previous works, as the first purely end-to-end automatic approach without human intervention. We select ASRF [15] and duration network [16], respectively, as representatives of the state-of-the-art baselines under the two above-mentioned branches in Section 5.

**Table 1.** State-of-the-art performance under different training conditions.

| Trained with Frame-Wise Annotations: | MPII Cooking2 [17] | MERL Shopping [11] | 50 Salads [18] | GETA [19] | JIGSAWS [20,21] |
|---|---|---|---|---|---|
| MSB-RNN [11] (2016) | 41.22% $(mAP)$ | 80.31% $(mAP)$ | | | |
| ED-TCN [12] (2017) | | 74.1% $(F.Acc)$ 64.2% $(mAP)$ | 64.7% $(F.Acc)$ ⋆ 59.8% $(Edit)$ ⋆ | 64.0% $(F.Acc)$ | 80.8% $(F.Acc)$ 84.7% $(Edit)$ |
| TCED [14] (2019) | | | 66.3% $(F.Acc)$ ⋆ 62.5% $(Edit)$ ⋆ 75.9% $(F.Acc)$ ⋆⋆ 71.3% $(Edit)$ ⋆⋆ | 63.4% $(F.Acc)$ 70.9% $(Edit)$ | 82.2% $(F.Acc)$ 87.7% $(Edit)$ |
| ASRF [15] (2021) | | | 84.5% $(F.Acc)$ ⋆ 79.3% $(Edit)$ ⋆ 84.9% $(F1@0.10)$ ⋆ 83.5% $(F1@0.25)$ ⋆ 77.3% $(F1@0.50)$ ⋆ | 77.3% $(F.Acc)$ 83.7% $(Edit)$ 89.4% $(F1@0.10)$ 87.8% $(F1@0.25)$ 79.8% $(F1@0.50)$ | |
| **Trained with Only Order of Actions:** | **Breakfast [22]** | **Hollywood Ext. [23]** | **50 Salads** | **YouTube Instructions [24]** | **YouCook2 [25]** |
| GRU + HMM [1] (2017) | 33.3% $(MoF)$ 47.3% $(IoD)$ ° | 11.9% $(IoU)$ 51.1% $(IoD)$ ° | | | |
| ProcNet [25] (2018) | | | | | 50.6% $(Jac.)$ °° 37.0% $(IoU)$ °° 37.1% $(Rec.)$ °° 30.4% $(Prec.)$ °° 33.4% $(F1)$ °° |
| Unsupervised [26] (2019) | 41.8% $(MoF$ °°$)$ | | 30.2% $(MoF$ °°⋆$)$ 35.5% $(MoF$ °°⋆⋆$)$ | 39.0% $(MoF$ °°$)$ | |
| Duration [16] (2021) | 55.7% $(F.Acc)$ 36.3% $(IoU)$ | 50.1% $(F.Acc)$ 31.4% $(IoU)$ | | | |

All results are reported without any form of supervision on test data, with only the temporal ordering of actions available for training, unless marked with °. ° denotes that additional information on the temporal ordering of actions is also provided in the test video. °° denotes even weaker supervision, where even the order of actions is not available, only un-ordered sets of the actions contained are available. All results are performed with low-level granularity with 48 sub-action classes, where each mid-level granularity is further divided into pre-, core-, and post-phases, unless marked with ⋆. ⋆ denotes that evaluations were performed at mid-level action granularity with 18 sub-action classes. The mid-level labels differentiate actions like `cut tomato` from `cut cucumber`, whereas the higher-level labels combine these into a single class, `cut`. ⋆⋆ denotes that evaluations were performed at eval-level action granularity with 9 sub-action classes such as `cut`, `peel`, and `add dressing`. Metrics definitions: (1) *MoF* (mean-over-frames) is equivalent to *F.Acc*, and refers to the average percentage of correctly labeled frames. (2) *IoU*, *IoD*, and *Jac.* (intersection over union/detection/prediction) refer to $\frac{I \cap I^*}{I \cup I^*}$, $\frac{I \cap I^*}{I}$, and $\frac{I \cap I^*}{I^*}$, respectively, where $I$ and $I^*$ are ground-truth and prediction intervals, respectively. *IoU* penalizes all the misalignment of proposal while *Jaccard* only penalizes the partition of segments beyond the ground truth. (3) *Edit*, *F1*, *Prec.*, and *Rec.*: see Equations (10)–(12) in Section 4.3 for definitions.

### 2.3. Automatic Feature Extraction

Automatic feature extraction is fundamental in all video recognition tasks. Early methods normally devise hand-crafted features [27,28]. The development of deep learning enables end-to-end automatic feature extraction, including two-stream network [29], 3D ConvNet (C3D) [30], and inflated 3D (I3D) [31,32]. We adopt I3D for feature extraction.

### 2.4. Automatic Label Extraction

Automatic label extraction is also related, as our action order comes from transcripts of video instructions [24,33]. Unlike these approaches focusing on the text processing part of the task, we assume that the discrete target label sequences are available in training stages.

### 3. The Proposed Encoder–Decoder Network
#### 3.1. Spatial Encoder

Generally, common 2D/3D backbones such as VGG [34], Residual [35], Inception [36], and I3D [31] variants can be applied orthogonally; in our empirical practice it is found that deep 3D backbones such as Res3D (Table 2 and Figure 5) generally yield better performance and efficiency. The convolutional feature maps of $T$ frame representations

$x_{1:T} = (x_1, x_2, \cdots, x_T)$ obtained from the last pooling layer (i.e., *pool*49) with size $L \times \frac{W}{32} \times \frac{H}{32}$ are fed into temporal encoding architectures.

**Table 2.** Network architecture of the I3D Res-50 backbone [37]. Residual blocks are shown in brackets, next to which is the number of repeated blocks in the stack. Each convolutional layer is followed by batch normalization and ReLU. Down-sampling is performed after each stack with a stride of 2 along dimensions of width and height. *res*5 culminates with a global spatiotemporal pooling layer outputting a 512-dimensional feature vector, which is subsequently fed to a fully connected layer outputting final class probabilities through softmax activations. The dimension *C* of the last fully connected layer is equal to the number of classes in the target dataset.

| Layer | I3D Res-50 | Output Size |
|:---:|:---:|:---:|
| *res*1 | $1 \times 7 \times 7, 64$ | $L \times \frac{W}{2} \times \frac{H}{2}$ |
| *res*2 | $\begin{bmatrix} 1 \times 1 \times 1,\ 64 \\ 1 \times 3 \times 3,\ 64 \\ 1 \times 1 \times 1,\ 256 \end{bmatrix} \times 3$ | $L \times \frac{W}{4} \times \frac{H}{4}$ |
| *res*3 | $\begin{bmatrix} 1 \times 1 \times 1,\ 128 \\ 1 \times 3 \times 3,\ 128 \\ 1 \times 1 \times 1,\ 512 \end{bmatrix} \times 4$ | $L \times \frac{W}{8} \times \frac{H}{8}$ |
| *res*4 | $\begin{bmatrix} 1 \times 1 \times 1,\ 256 \\ 1 \times 3 \times 3,\ 256 \\ 1 \times 1 \times 1,\ 024 \end{bmatrix} \times 6$ | $L \times \frac{W}{16} \times \frac{H}{16}$ |
| *res*5 | $\begin{bmatrix} 1 \times 1 \times 1,\ 512 \\ 1 \times 3 \times 3,\ 512 \\ 1 \times 1 \times 1,\ 2048 \end{bmatrix} \times 3$ | $L \times \frac{W}{32} \times \frac{H}{32}$ |
| *fc* | global pooling $\rightarrow$ softmax | $1 \times 1 \times C$ |

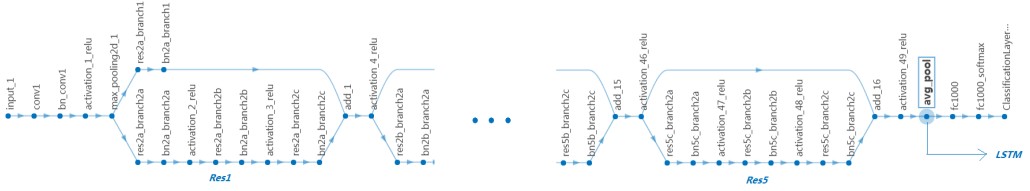

**Figure 5.** Res-50 backbone, 2nd, 3rd, 4th residual blocks are internally similar and are therefore omitted.

### 3.2. Temporal Encoder

In this section, we formally describe three common temporal architectures Table 3, and empirically compare them for this task in Section 5, in terms of performance accuracy, training time, generalization at test time, and ease of use.

**Table 3.** Structural and training details of the temporal backbones used in experiments.

| | BLSTM | Full Convolution | Transformer |
|:---|:---:|:---:|:---:|
| Dimension | $2 \times 1024$ | $15 \times (Xeption\_Block + 1536)$ | $(6 + 6) \times (512 + 2048 + 512)$ |
| # Parameters | 22 M | 35 M | 40 M |
| Optimization | SGD, batch size $= 1$ | Adam, batch size $= 32$ | Adam, batch size $= 32$ |
| Momentum | 0.9 | − | − |
| Learning Rate | $10^{-2} \rightarrow 10^{-4}$ | $10^{-3} \rightarrow 10^{-4}$ | $10^{-3} \rightarrow 10^{-4}$ |
| Dropout | 0.3 | 0.8 | 0.1 |
| Training Time/Batch | 0.76 s/32 | 0.34 s | 0.41 s |
| ⋆ Convergence (Iterations) | $1.0 \times 10^6 \times 32$ | $0.8 \times 10^6$ | $3.0 \times 10^6$ |
| ⋆ Convergence (Clock Time) | 9 days | 3.4 days | 14 days |

⋆ Statistics are collected on the Breakfast dataset [38] on a single GPU; however, convergence curves are generally consistent across the other datasets (Section 4.2) and are therefore omitted for presentation. The time to train the I3D Res-50 CNN (2 weeks) is also excluded from the statistics.

### 3.2.1. LSTM

Following [39], we adapt a 1-layer bidirectional LSTM (BLSTM), taking the vision feature vectors $x_t$ as input and outputting a class probability $y_t(\pi_t)$, $\pi_t \in \mathcal{A}$ for every frame $t$. The BLSTM has 1024 cells in each direction. The overall network is trained together with CTC (see Section 3.4.2 for training detail). The output alphabet $\mathcal{A}$ is therefore augmented with the CTC blank class label, and the decoding is performed with a beam search.

### 3.2.2. Full Convolution

Following [40], we adapt depth-wise separable convolution layers, which consist of a separate convolution along the time dimension for every channel, followed by a projection along the channel dimensions (a position-wise convolution with filter width 1). Each spatial convolution is followed by a shortcut connection, batch normalization, and ReLU. The overall network consists of 15 convolutional layers, also trained with a CTC loss, with sequences decoded by using a beam search (above).

### 3.2.3. Transformer

Following [41], we adopt 6 encoder and 6 decoder layers, $\log_2 |\mathcal{A}|$ attention heads, and each attention has 512 channels and is followed by two position-wise fully connected layers with 2048 and 512, compared to that of 1536 for the fully convolutional network. Every encoder layer is a self-attention, where the input tensor serves as the attention queries, keys, and values at the same time.

Every decoder layer attends on the embedding produced by the encoder using common soft-attention: the encoder outputs are the attention keys and values, and the previous decoding layer outputs are the queries. The decoder produces target class probabilities which are matched to the ground-truth labels by CTC decoding and trained as a whole with a cross-entropy loss.

This section demonstrates how to achieve more efficient decoding by the early elimination of paths that obviously violate the visual consistency based on existing labels. To understand how to reduce the search space of eligible candidate decoding paths, it is necessary that we first briefly look back on how the original CTC decoding process performs general end-to-end alignment learning in sequential signal transduction tasks.

### 3.3. The Original CTC Decoding

CTC sums over all possible alignments upon conditional independent assumptions. Firstly, given a training sample of $T$ frames:

$$\boldsymbol{x} = (x_1, \cdots, x_T) \tag{1}$$

where $t$ is the index of $T$ frames and $x_t$ is a vector of frame features.

According to the conditional independence assumption (CIA) from the original CTC formulation [7], the probability of path $\boldsymbol{\pi} = (\pi_1, \cdots, \pi_T)$, $\pi_t \in \mathcal{A}$ is the stepwise product of the network output softmax activation $y_t(\pi_t)$ of $\pi_t$ at each frame $t$:

$$P(\boldsymbol{\pi}|\boldsymbol{x}) \overset{\text{CIA}}{=\!=\!=} \prod_{t=1}^{T} y_t(\pi_t) \tag{2}$$

where $y_t(k)$ is the probability of the network outputting action $k$ at time $t$, given input $\boldsymbol{x}$, $k \in \mathcal{A}$ ($\mathcal{A}$ is the collective set of all possible actions).

We refer to $\boldsymbol{\pi}$ over $\mathcal{A}$ as *paths*, to be distinguished from the action order $\boldsymbol{l}$, which is naturally produced from path $\boldsymbol{\pi}$ by applying the operator that removes repetitions $\mathcal{B}(\boldsymbol{\pi})$:

$$\mathcal{B}(\boldsymbol{\pi}) = \boldsymbol{l} \tag{3}$$

The probability of $\boldsymbol{l}$ sums over all paths consistent with $\boldsymbol{l}$:

$$P(\boldsymbol{l}|\boldsymbol{x}) = \sum_{\boldsymbol{\pi} \in \mathcal{B}^{-1}(\boldsymbol{l})} P(\boldsymbol{\pi}|\boldsymbol{x}) \overset{\text{Equation (2)}}{=\!=\!=\!=} \sum_{\mathcal{B}(\boldsymbol{\pi})=\boldsymbol{l}} \prod_{t=1}^{T} y_t(\pi_t) \tag{4}$$

Finally, the negative log likelihood of observing $\boldsymbol{l}$:

$$\mathcal{J} = -\ln P(\boldsymbol{l}|\boldsymbol{x}) \tag{5}$$

### 3.4. Decoding Search Space Reduction

One drawback of original CTC is that Equation (2) weights all paths equally, causing the sum in Equation (4) to include visually inconsistent paths $\pi$ that deteriorate the performance.

We incorporate visual similarity into by Equation (2) rewarding paths:

$$P(\boldsymbol{\pi}|\boldsymbol{x}) \propto \prod_{t=1}^{T} \phi_t \cdot \psi_t^{t+1}$$

$$\text{where:} \quad \phi_t = y_t(\pi_t),$$

$$\psi_t^{t+1} = \begin{cases} \max(\theta, s_t^{t+1}) & \pi_t = \pi_{t+1} \\ \theta & \pi_t \neq \pi_{t+1} \end{cases} \qquad (6)$$

$$s_t^{t+1} = f_{sim}(x_t, x_{t+1})$$

where $\phi_t = y_t(\pi_t)$ represents the original Equation (2) formulation, and $\theta$ is a minimum threshold of the frame similarity function $s_t^{t+1} = f_{sim}(x_t, x_{t+1})$.

- When $\pi_t = \pi_{t+1}$ and $s_t^{t+1} > \theta$ (high similarity), $\psi_t^{t+1} = s_t^{t+1}$ reward path to stay at the same prediction.
- When $\pi_t = \pi_{t+1}$ and $s_t^{t+1} < \theta$ (low similarity), $\psi_t^{t+1} = \theta$ means no intervention after normalization.

### 3.4.1. Example

Figure 6 illustrates how Equation (6) re-weights a ground-truth path and a degenerated path. In Figure 6, there is a ground-truth path $\pi^1$ and another path $\pi^2$ that produces the same action order with $\pi^1$ $\left(\mathcal{B}(\pi^1) = \mathcal{B}(\pi^2) = \ell\right)$, but yields different frame-wise label sequences:

$\mathcal{A} = \{Action1, Action2, Action3\}$, represented by green, yellow, and orange nodes, respectively.

$\ell = green \rightarrow yellow \rightarrow orange$, which is supposed to be already known during training.

$\pi^1 = green \rightarrow green \rightarrow green \rightarrow yellow \rightarrow yellow \rightarrow orange$.

$\pi^2 = green \rightarrow yellow \rightarrow orange \rightarrow orange \rightarrow orange \rightarrow orange$.

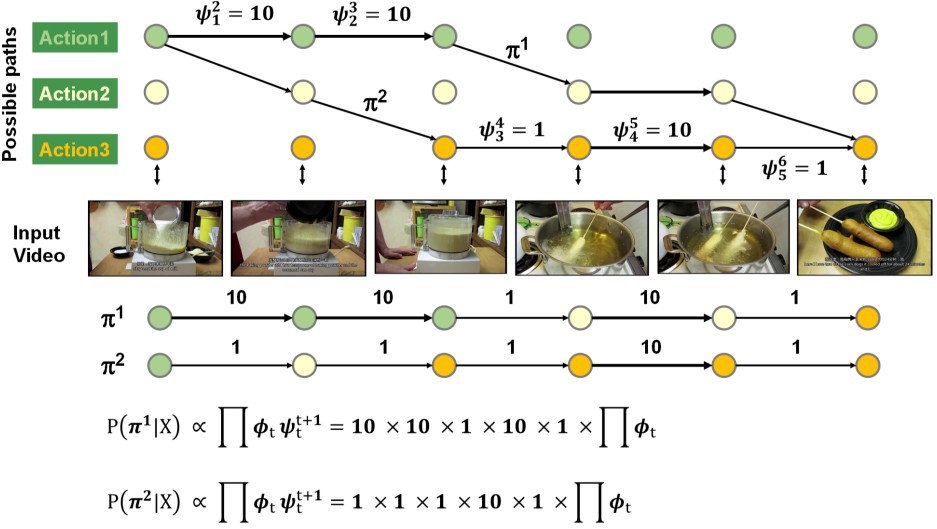

**Figure 6.** For illustration, an input with $T = 6$ frames and $|\mathcal{A}| = 3$ annotated actions. High similarity is indicated by thick lines between frames. Note that $\pi_1$ stays at the same action prediction across similar frames. Consequently, Equation (6) weights $\pi_1$ higher over $\pi_2$. In contrast, $\pi_1$ and $\pi_2$ are equally treated in original CTC.

In fact, for $T = 6$, $|\boldsymbol{\ell}| = 3$, there are altogether $\begin{pmatrix} T - 2 \\ |\boldsymbol{\ell}| - 1 \end{pmatrix} = \begin{pmatrix} 4 \\ 2 \end{pmatrix} = \frac{4!}{2!2!} = 6$ distinct paths $\boldsymbol{\pi}^i$ satisfying the supervised constraints $\mathcal{B}(\boldsymbol{\pi}^i) = \boldsymbol{\ell}$, because:

- $\pi_1$ has to be $\boldsymbol{\ell}(1) = green$;
- $\pi_T$ has to be $\boldsymbol{\ell}(end) = orange$; and
- The difference among different $\boldsymbol{\pi}^i$s is to choose at which $\pi_t, t = 2, \cdots, T - 1$ to transit the node label from $\boldsymbol{\ell}(j)$ to $\boldsymbol{\ell}(j + 1)$, with $j = 1, 2, \cdots, |\boldsymbol{\ell}| - 1$.

To summarize, the necessary and sufficient condition for path $\boldsymbol{\pi}$ to be consistent with the supervised action order $\boldsymbol{\ell}$, if and only if:

- $\pi_1 = \boldsymbol{\ell}(1)$.
- $\pi_T = \boldsymbol{\ell}(end)$.
- For each middle node $\pi_t, t = 2, \cdots, T - 1$, there are only two possible options: (1) Stay the same as the previous node, which means if $\pi_{t-1} = \boldsymbol{\ell}(j)$, $\pi_t = \boldsymbol{\ell}(j)$ as well. Whenever this case holds true, a 'repetition' happens in $\boldsymbol{\pi}$. (2) Transit from $\pi_{t-1} = \boldsymbol{\ell}(j)$ to the next label in $\boldsymbol{\ell}$, which means $\pi_t = \boldsymbol{\ell}(j + 1)$. Any other label assignment will cause $\mathcal{B}(\boldsymbol{\pi}) = \boldsymbol{\ell}$ holding false.

So far we can draw the conclusion that the only difference between valid paths $\boldsymbol{\pi}$ and supervised action order $\boldsymbol{\ell}$ is that $\boldsymbol{\pi}$ contains 'repetitions'; by inserting and removing 'repetitions', $\boldsymbol{\pi}$ and $\boldsymbol{\ell}$ can be converted to each other.

In this example, $\pi_t = \pi_{t+1}$ holds true at $t = 1, 2, 4$ in path $\boldsymbol{\pi}^1$, and $t = 3, 4, 5$ in path $\boldsymbol{\pi}^2$. $\pi_t = \pi_{t+1}$ means that a 'repetition' happens in path $\boldsymbol{\pi}$ at frame $t + 1$, but there are two cases to judge whether such a 'repetition' should be encouraged or not if taking into account its consistency with the ground truth alignment.

Case 1: $s_t^{t+1} > \theta$,

which means that apart from the supervised order information $\boldsymbol{\ell}$, it is also unsupervisedly observed that frame $t + 1$ is visually in great similarity with frame $t$, which suggests such a 'repetition' should be additionally encouraged.

At $t = 1$, $\pi_t^1 = \pi_{t+1}^1 = green$ holds true for $\boldsymbol{\pi}^1$ and false for $\boldsymbol{\pi}^2$, where $\pi_t^2 = green \neq \pi_{t+1}^2 = yellow$. $\boldsymbol{\pi}^1$ is consistent with ground truth while $\boldsymbol{\pi}^2$ is not, since $s_t^{t+1} = 10 > \theta$. Therefore:

- $\psi_t^{t+1} = s_t^{t+1}$ for path $\boldsymbol{\pi}^1$ when $s_t^{t+1} > \theta$ is introduced to encourage $\boldsymbol{\pi}^1$, which has a 'repetition' at $t + 1 = 2$ to yield a higher probability than $\boldsymbol{\pi}^2$ after $P(\boldsymbol{\pi}|\boldsymbol{x})$ re-normalization at time step $t + 1$.
- $\psi_t^{t+1} = \theta$ for path $\boldsymbol{\pi}^1$ since $\pi_t^2 \neq \pi_{t+1}^2$ at $t = 1$, which means no encouragement after $P(\boldsymbol{\pi}|\boldsymbol{x})$ re-normalization at time step $t + 1$; the calculation remains the same as the original Equation (3).

Case 2: $s_t^{t+1} \leq \theta$,

which means that apart from the supervised order information $\boldsymbol{\ell}$, it is also unsupervisedly observed that frame $t + 1$ is visually not similar with frame $t$, which suggests such a 'repetition' should not be encouraged.

At $t = 3$, $\pi_t^2 = \pi_{t+1}^2 = orange$ holds true for $\boldsymbol{\pi}^2$ and false for $\boldsymbol{\pi}^1$, where $\pi_t^1 = green \neq \pi_{t+1}^1 = yellow$. $\boldsymbol{\pi}^1$ is consistent with ground truth while $\boldsymbol{\pi}^2$ is not, since $s_t^{t+1} = 1 \leq \theta$. Therefore:

- $\psi_t^{t+1} = \theta$ for path $\boldsymbol{\pi}^2$ when $s_t^{t+1} \leq \theta$, which means such a 'repetition' at $t + 1 = 4$ is not encouraged and $P(\boldsymbol{\pi}|\boldsymbol{x})$ remains the same as the original Equation (3) after re-normalization.
- $\psi_t^{t+1} = \theta$ for path $\boldsymbol{\pi}^1$ since $\pi_t^1 \neq \pi_{t+1}^1$ at $t = 3$, which also means no intervention.

### 3.4.2. How to Train the Proposed Decoder

The back-propagation through the CTC layer to obtain the closed form of $\frac{\partial y_t(k*)}{\partial a_t(k)}$ is quite cumbersome, and may last for several pages, so we chose not to present the mathematical derivation in too much detail; for readers interested in the complete derivation of Equation (7) to obtain the gradient of $P(\boldsymbol{\ell}|x)$, it can be easily found in the relevant literature or tutorials, such as [7,42], based on dynamic programming under chained rule of derivation.

Here we briefly give the closed form for forward loss function calculation $\mathcal{J} = -\ln P(\boldsymbol{\ell}|x)$, together with its backward gradient w.r.t. the neural network output $y_t(k)$ (the response of label $k$ at time $t$):

$$P(\boldsymbol{\ell}|x) = \sum_{j=1}^{J=|\boldsymbol{\ell}|} \alpha_t(j) \cdot \beta_t(j) \tag{7}$$

$$\frac{\partial \mathcal{J}}{\partial y_t(k)} = -\frac{\partial \ln P(\boldsymbol{\ell}|x)}{\partial y_t(k)} = -\frac{1}{P(\boldsymbol{\ell}|x)} \frac{\partial P(\boldsymbol{\ell}|x)}{\partial y_t(k)}$$

$$= -\frac{1}{P(\boldsymbol{\ell}|x) \cdot y_t(k)} \sum_{j \in label(k)} \alpha_t(j) \cdot \beta_t(j)$$

$$\frac{\partial \mathcal{J}}{\partial a_t(k)} = \sum_{k*} \frac{\partial \mathcal{J}}{\partial y_t(k*)} \cdot \frac{\partial y_t(k*)}{\partial a_t(k)}$$

$$= y_t(k) - \frac{1}{P(\boldsymbol{\ell}|x)} \left( \sum_{j \in label(k)} \alpha_t(j) \cdot \beta_t(j) \right)$$

where:

$a_t(k)$ are the un-normalised outputs before the softmax activation function is applied: $y_t(k) = \frac{\exp(a_t(k))}{\sum_{k*} \exp(a_t(k*))}$, $k*$ ranges over all outputs; and

$\alpha_t(j)$, $\beta_t(j)$ are *forward* and *backward* variables, respectively. $\alpha_t(j)$ is defined as the summed probability of all paths satisfying $\mathcal{B}(\boldsymbol{\pi}_{1:t}) = \boldsymbol{l}_{1:j}$, and $\beta_t(j)$ appends to $\alpha_t(j)$ from $t+1$ that completes $\boldsymbol{l}$, where $\boldsymbol{l}_{1:j}$ is the first $j$ actions of $\boldsymbol{\ell}$; both $\alpha_t(j)$ and $\beta_t(j)$ can be calculated by recursive inductions:

$$\alpha_t(j) = \alpha_{t-1}(j)y_{t-1}(b) + \alpha_{t-1}(j-1)y_{t-1}(j) \tag{8}$$

$$\beta_t(j) = \beta_{t+1}(j)y_t(b) + \beta_{t+1}(j+1)y_t(j+1)$$

$$\alpha_1(j) = \begin{cases} \alpha_{t=1}(j=1) = 1 \\ \alpha_{t=1}(j \neq 1) = 0 \end{cases}$$

$$\beta_T(j) = \begin{cases} \beta_{t=T}(j = |\boldsymbol{\ell}|) = 1 \\ \beta_{t=T}(j \neq |\boldsymbol{\ell}|) = 0 \end{cases}$$

$$j = 1, \cdots, |\boldsymbol{\ell}|$$

$\frac{\partial y_t(k*)}{\partial a_t(k)}$ in Equation (7) is the final 'error signal' back-propagated through the network during training.

## 4. Experimental Setup

### 4.1. Visual Similarity Measurement

Cut the video into $T/M$ clusters, each with length $M$. Set $M$ to be conservatively shorter than common action lengths (e.g., ~400 frames on average in the YouCook2 dataset) so that frames belonging to different actions do not blend into the same cluster.

Thus, $s_t^{t+1}$ can be set under the resulting constraints that:

- $\pi_t = \pi_{t+1}$ if and only if $x_t$ and $x_{t+1}$ fall within same cluster;
- $\pi_t \neq \pi_{t+1}$ if and only if $x_t$ or $x_{t+1}$ is at the boundary between clusters.

$$s_t^{t+1} = f_{sim}(x_t, x_{t+1}) = \begin{cases} \infty & \pi_t = \pi_{t+1} \\ \cos(x_t, x_{t+1}) = \frac{x_t \cdot x_{t+1}}{|x_t||x_{t+1}|} & \pi_t \neq \pi_{t+1} \end{cases} \tag{9}$$

### 4.2. Datasets

We evaluate the proposed approach on four public available datasets: YouTube Instructional (https://www.di.ens.fr/willow/research/instructionvideos, accessed on 12 January 2021), YouCook2 (http://youcook2.eecs.umich.edu, accessed on 12 January 2021), Breakfast (https://serre-lab.clps.brown.edu/resource/breakfast-actions-dataset, accessed on 12 January 2021), and 50 Salads (http://cvip.computing.dundee.ac.uk/datasets/foodpreparation/50salads, accessed on 12 January 2021).

**YouTube Instructions [24]**  contains 150 samples from YouTube on five tasks: making coffee, changing a car tire, CPR, jumping a car, and potting a plant, with approximately two minutes per sample.

**YouCook2 [25]**  contains about 2$k$ samples from YouTube on 90 cooking recipes, with approximately 3∼15 steps per recipe class, where each step is a temporally aligned narration collected from paid human workers.

**Breakfast [38]**  contains about 2$k$ samples on ten common kitchen tasks with approximately eight steps per task. The average length of each task varies from 30 s to 5 min.

**50 Salads [18]**  contains 4.5 h of 25 people preparing 2 mixed salads each, with approximately 10$k$ frames per sample. Each sub-action corresponds to two levels of granularity, and each low-level granularity is further divided into pre-, core-, and post-phase.

### 4.3. Metrics

**Frame-level accuracy**  is calculated as the percentage of correct predictions. Intuitively, frame-wise metrics ignore temporal patterns and occurrence orders in the sequential inputs. It is possible to achieve high frame measures but at the same time generate considerable over-segmentation errors, as visualized later, raising the need to introduce segmental metrics to penalize predictions that are out of order or over-segmented.

**Segment-level edit distance**  is also known as the Levenshtein distance, and only measures the temporal order of occurrence, without considering durations. It is therefore useful for procedural tasks in this work, where the order is the most essential.

It is calculated as segment insertions, deletions, and substitutions between predicted order and the ground-truth sequence, then normalized to range [0∼100] in Table 4 such that higher is better:

$$Edit = \frac{|insertions| + |deletions| + |substitutions|}{|Ground\ truth|} \tag{10}$$

**Table 4.** Accuracy across different datasets (%). All referenced baselines under both training conditions (Row 1∼4) are re-implemented under the same empirical control (i.e., identical datasets and evaluation metrics) to facilitate consistent comparisons against our proposed variants *. Results are means over 4 runs (variances are omitted for readability). All differences are significant ($p < 0.01$). The bold-faced font highlights the best results obtained on the given data set. The values in brackets calculate the absolute increments of the current row (denoted by ◯) relative to the row referenced in the rightmost column. The second to the rightmost column calculates the statistical means and variances of the corresponding increments across different datasets within the same row.

| Datasets | | | (1) 50 Salads [18] (2013) | YouTube Instructions [24] (2016) | Breakfast [38] (2016) | YouCook2 [25] (2018) | |
|---|---|---|---|---|---|---|---|
| | Size (hour) | | 4.5 h | 5 h | 77 h | 176 h | |
| | #Samples | | 50 | 150 | 2k | 2k/14k | |
| | #Classes/#Sub-actions | | 2/17 | 5/− | 10/62 | 90/open | |
| | Sample Length | | 4∼5 min | 2 min | 0.5∼5 min | 5 min | |
| | Label Length | | 37∼72 steps | 7∼10 steps | 3∼10 steps | 3∼15 steps | |
| | Recording ($width \times height \times fps$) | | $640 \times 480 \times 30$ | YouTube | $320 \times 240 \times 15$ | YouTube | |
| | Recording ($Camera/Background$) | | Fixed/Stable | Dynamic/Open | Fixed/Stable | Dynamic/Open | |
| **Frame-wise Annotation** | 1. CNN + BLSTM [11] (2016) | Frame Acc. | 76.9 | 60.8 | 60.6 | 45.8 | |
| | | mAP @ 0.25 | 72.5 | 47.0 | 64.9 | 33.7 | |
| | | Edit | 71.4 | 41.9 | 61.8 | 28.4 | |
| | 2. ED-TCN [12] (2017) | Frame Acc. | 82.1 | 64.9 | 64.7 | 48.9 | |
| | | mAP @ 0.25 | 73.4 | 50.2 | 65.7 | 36.5 | |
| | | Edit | 68.9 | 44.7 | 59.6 | 30.4 | |
| | 3. TCED [14] (2019) | Frame Acc. | 68.1 | 66.0 | 53.7 | 47.9 | |
| | | mAP @ 0.25 | 68.5 | 54.0 | 61.3 | 35.9 | |
| | | Edit | 66.0 | 48.3 | 57.1 | 30.7 | |
| | 4. ASRF [15] (2021) | Frame Acc. | **84.5 (+4.5)** | **81.9 (+5.5)** | **67.6 (+4.5)** | **59.5 (+11.8)** | (8.05 ± 1.90) |
| | | mAP @ 0.25 | **83.5 (+11.9)** | **65.8 (+4.4)** | **72.4 (+8.3)** | **43.8 (+8.1)** | (9.53 ± 4.44) |
| | | Edit | **79.3 (+4.1)** | 58.0 (−2.5) | **68.9 (+3.8)** | **36.9 (+2.7)** | (8.23 ± 2.32) |
| **Video-level Annotation** | 5. Duration [16] (2021) (*Baseline*) | Frame Acc. | 70.7 | 67.5 | 55.7 | 42.1 | $\Delta_{Base} =$ (◯ − 5. Baseline) |
| | | mAP @ 0.25 | 63.2 | 54.2 | 56.6 | 31.5 | |
| | | Edit | 59.3 | 47.8 | 51.3 | 26.9 | |
| | 6. I3D + BLSTM + CTC (Ours) | Frame Acc. | 56.8 (−13.8) | 54.3 (−13.2) | 44.8 (−10.9) | 33.8 (−8.2) | (−11.54 ± 2.53) |
| | | mAP @ 0.25 | 50.8 (−12.3) | 43.6 (−10.6) | 45.5 (−11.1) | 25.3 (−6.1) | (−10.05 ± 2.69) |
| | | Edit | 47.7 (−11.6) | 38.4 (−9.3) | 41.3 (−10.0) | 21.7 (−5.3) | (−9.07 ± 2.69) |
| | 7. I3D + FC + CTC (Ours) | Frame Acc. | 62.9 (−7.7) | 60.1 (−7.4) | 49.6 (−6.1) | 37.5 (−4.6) | (−6.46 ± 1.42) |
| | | mAP @ 0.25 | 56.5 (−6.6) | 48.5 (−5.7) | 50.6 (−5.9) | 28.2 (−3.3) | (−5.41 ± 1.45) |
| | | Edit | 59.6 (+0.33) | 48.1 (+0.26) | 51.6 (+0.29) | 27.1 (+0.15) | (0.26 ± 0.07) |
| | 8. I3D + Transformer + CTC (Ours) | Frame Acc. | 74.9 (+4.3) | 71.6 (+4.1) | 59.1 (+3.4) | 44.6 (+2.6) | (3.58 ± 0.78) |
| | | mAP @ 0.25 | 67.1 (+3.9) | 57.6 (+3.4) | 60.1 (+3.5) | 33.5 (+1.9) | (3.20 ± 0.85) |
| | | Edit | 70.6 (+11.3) | 56.9 (+9.1) | 61.1 (+9.8) | 32.1 (+5.1) | (8.83 ± 2.62) |
| | 9. I3D + Transformer + R-CTC (Ours) | Frame Acc. | 79.9 (+9.3) | 76.3 (+8.8) | 63.0 (+7.3) | 47.6 (+5.5) | (7.74 ± 1.70) |
| | | mAP @ 0.25 | 71.5 (+8.3) | 61.3 (+7.1) | 64.0 (+7.4) | 35.6 (+4.1) | (6.76 ± 1.81) |
| | | Edit | 75.1 (+15.8) | **60.5 (+12.7)** | 65.0 (+13.7) | 34.1 (+7.2) | (12.3 ± 3.68) |

[1] Evaluations on 50 salads were performed at mid-level action granularity with 18 sub-action classes. * Some values reported in this table may differ from the values in the original literature even under the same dataset/training/evaluation combinations due to the re-implementation; if readers are interested in the performance originally reported, please check their citation link in the referenced methods.

**Segment-level mean average precision (*mAP@k*)** With an intersection over union (IoU) threshold $k$, calculated as dividing the intersection between each pair of predicted segments $I$ and the ground-truth segment of the same action category $I^*$ by their union:

$$IoU(I) = \frac{|I \cap I^*|}{|I \cup I^*|} \tag{11}$$

$I$ is considered as a 'true positive' ($TP$) if $IoU(I) \geq k$, otherwise it is a 'false positive' ($FP$). Average precision is accumulated across all categories. $mAP@k$ is more invariant to small temporal shifts as compared to the above metric.

**Segment-level F1-score (*F1@k*)** With an intersection over union (IoU) threshold $k$, where true positives are judged by $IoU(I) \geq k$ with labels same as the ground truth:

$$
\begin{aligned}
Precision &= \frac{TP}{TP + FP} \\
Recall &= \frac{TP}{TP + FN} \\
F1 &= 2 * \frac{Recall \cdot Precision}{Recall + Precision}
\end{aligned}
\tag{12}
$$

*4.4. Hyper-Parameters*

**CNN pre-training:** I3D Res-50 backbone [37] pre-trained on Kinetics [43] (https://github.com/deepmind/kinetics-i3d, accessed on 20 August 2020 ).

**Optimization:** The BLSTM is trained with Vanilla SGD, a fixed momentum of 0.9, initial learning rate $10^{-2}$ and reduced down to $10^{-4}$ every time the error plateaus. Following [44], the fully convolutional network and Transformer are trained with the ADAM optimizer, initial learning rate $10^{-3}$, reduced down to $10^{-4}$ on plateaus.

**Transformer embedding:** The information about the sequence order of the encoder and decoder inputs is fed to the model via fixed positional embedding in the form of sinusoid functions. The Transformer is trained using teacher forcing—the ground truth of the previous decoding step is fed as the input to the decoder, while during inference the decoder prediction is fed back.

**Dropout:** Following [45], the BLSTM is trained with dropout probability $p = 0.3$ on the units of the inputs and recurrent layers. The fully convolutional network is trained with a dropout probability $p = 0.8$ on the units of batch normalization layers. The Transformer is trained with dropout probability dropout $p = 0.1$.

**Termination:** The loss function no longer drops on the validation set between 2 consecutive epochs.

**Software and Hardware:** All the models are implemented in TensorFlow and trained on a single GeForce GTX 1080 Ti GPU with 11 GB memory.

**5. Results and Discussion**

*5.1. Ablation Analysis*

Rows 5∼9 of Table 4 show the ablation studies of combining different modules elaborated in Section 3. The final performance improvement can be estimated by the average differences across the four tested benchmarks between Baseline (Row 5. Baseline) and the best-performing 9. I3D + Transformer + R-CTC, which is denoted as $\Delta_{Base}$ hereafter for simplicity:

$$
\Delta_{Base}(9) = (\overline{\Delta_{F.Acc}} = 7.74 \pm 1.70, \overline{\Delta_{mAP}} = 6.76 \pm 1.81, \overline{\Delta_{Edit}} = 12.3 \pm 3.68)
$$

5.1.1. Ablation Analysis of Temporal Encoding Backbone

Rows 6∼8 investigate the impact of different temporal backbones (Section 3) separately. The average differences between BLSTM, the fully convolutional network, and Transformer under original CTC across the four tested benchmarks were:

$$
\Delta_{Base}(6) = (\overline{\Delta_{F.Acc}} = -11.54 \pm 2.53, \overline{\Delta_{mAP}} = -10.05 \pm 2.69, \overline{\Delta_{Edit}} = -9.07 \pm 2.69)
$$

$$
\Delta_{Base}(7) = (\overline{\Delta_{F.Acc}} = -6.46 \pm 1.42, \overline{\Delta_{mAP}} = -5.41 \pm 1.45, \overline{\Delta_{Edit}} = 0.26 \pm 0.07)
$$

$$
\Delta_{Base}(8) = (\overline{\Delta_{F.Acc}} = 3.58 \pm 0.78, \overline{\Delta_{mAP}} = 3.20 \pm 0.85, \overline{\Delta_{Edit}} = 8.83 \pm 2.62)
$$

LSTM performed worse than the fully convolutional network, even though the recurrent model has full context of every decoding time step compared to the convolutional network that only looks at a limited time window of the input. This is in accordance with current trends shifting from recurrent networks towards purely convolutional/self-attentional models in other related domains, such as translation [46,47] and speech [48,49]. Dedicated explorations [50,51] blame this inferiority to the inherent limitations of recurrent structures. Although the gating mechanism alleviates the difficulty of gradient propagation, the maximum memory is still restricted to a limited distance, usually not exceeding $\Theta(10^2)$ time steps. The fully convolutional model has a smaller number of parameters and trains faster than BLSTM and bidirectional Transformer; the best-performing backbone was the Transformer.

Training Time

Transformer and the fully convolutional network took approximately the same amount of time to compute a batch of 32 samples (Table 3). This is in accordance with the theoretical complexity ($\Theta(Td^2)$) for each layer of both models, where $d$ is the dimension of channels. Transformer has fewer channels (512) for every self-attention block, but it is in effect a deeper model, with $3 \times (6 + 6) = 24$ layers in total. However, the fully convolutional network took fewer iterations to converge, completing the full curriculum in 3.5 days, compared to the 14 days for Transformer. This may be due to the more complex contraction among the self-attention queries, keys, and values during the gradient propagation and weight updating. In contrast, the fully convolutional model has no reverse connections among learnable modules given a fixed context. The BLSTM naturally took more time to run one batch, since the computations within its layers have to be executed sequentially; consequently it converged in almost double the clock time of the fully convolutional network.

5.1.2. Ablation Analysis of CTC Search Space Reduction

Row 8 and 9 investigate the impact of the decoding block (Section 3.4) separately. The average differences between the proposed R-CTC and the standard original CTC across the four tested benchmarks were:

$$\Delta_{Decoding}(9 - 8) = (\overline{\Delta_{F.Acc}} = 9.8 \pm 1.25, \overline{\Delta_{mAP}} = 6.53 \pm 2.29, \overline{\Delta_{Edit}} = 4.23 \pm 1.58)$$

Generally, original CTC performed quite competitively at the order level, despite its rather poor performance at frame-level (the finer the evaluation metric applied, the larger the gap grew in the rightmost column of Rows 8 and 9, highlighted as brown background cell color, growing darker as the cell value grows greater), which is consistent with its underlying principle to learn order distributions rather than alignments. The performance improvement was statistically significant in all 12 dataset/measurement combinations tested, indicating that the proposed decoding procedure is effective.

Training Time

Figure 7 compares learning efficiency. The non-CTC model converged the fastest but the final loss on validation was the highest. CTC converged slower than non-CTC baseline because of the increased depth in the gradient propagation process. Although the gradient depth was further increased in reduced-CTC compared to the original CTC, its convergence process did not deteriorate but even accelerated, thanks to the significantly reduced search space of potential alignments.

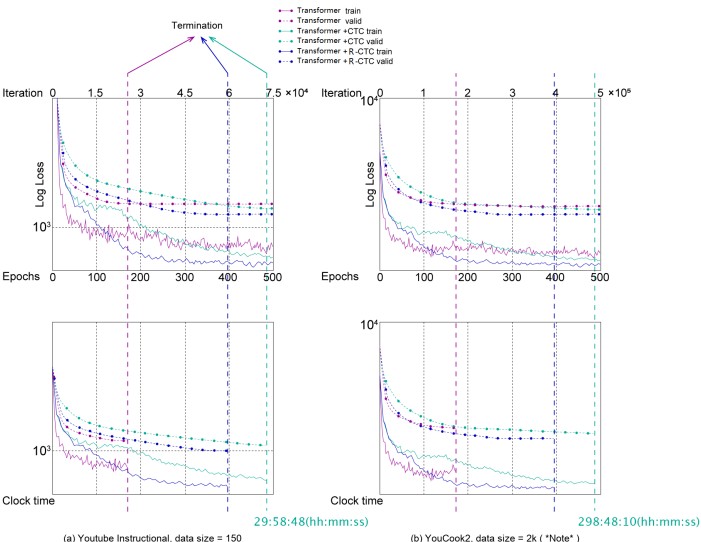

**Figure 7.** Loss curves in log-scale w.r.t. the number of epochs and elapsed clock time. Vertical dotted lines of different colors indicate that the loss function on the corresponding validation no longer dropped between two consecutive epochs, which is the position where the virtual learning process actually terminated. (Note: Due to time and hardware limitations, only a subset of $\sim 1k$ samples were selected for cross-validation training on the YouCook2 dataset).

## 5.2. Comparison with State-of-the-Art Frame-Wise Methods

In contrast to video-level supervision, fully supervised segmentation is a much more researched topic, and there are more diverse benchmarks available for comparison. As expected, there was an obvious gap between the best-performing video-level methods and the state-of-the-art frame-wise method (4. ASRF):

$$\Delta_{Frame}(4-9) = (\overline{\Delta_{F.Acc}} = 8.05 \pm 1.90, \overline{\Delta_{mAP}} = 9.53 \pm 4.44, \overline{\Delta_{Edit}} = 8.23 \pm 2.32)$$

Although 11 out of the 12 current best results obtained in different dataset/measurement combinations were provided by fully supervised approaches (highlighted in bold font in Table 4), they rely on expensive frame-by-frame manual annotations and task-specific hand-engineered pre-/post-processing techniques, while our proposal is purely automatic end-to-end without human intervention, both during the training phase and after deployment. Note that on the second dataset (YouTube Instructions), our proposed 9. I3D + Transformer + R-CTC even outperformed the best frame-wise result by a noticeable margin in terms of order-level evaluation (+12.7% by re-normalized *Edit* measurement). The performance improvement was statistically significant, indicating the possibility that the explicit alignment learning in a purely data-driven fashion may be more effective than fully supervised methods if end-to-end ordering information is well-learned.

## 5.3. Qualitative Analysis

Some representative examples are shown in Figure 8. Baseline without CTC outputs were more noisy and over-segmented actions. CTC outputted a degenerated path, however the order was correct. Reduced-CTC did not suffer from over-segmentation, and had better localization and ordering.

Specifically, we found that Transformer + RCTC captured longer-range temporal dependencies than the state of the art, especially in cases when distinct actions were visually very similar. For example, Baseline wrongly predicted the ground-truth class 'add coffee' as 'pour coffee', while neglecting the temporal dependencies between certain action pairs. Our hybrid encoder also made more reliable predictions of extremely short action instances that fell in between two long actions ('butter pan', 'withdraw stove', etc.). This suggests that using modified temporal classification is critical for improving the accuracy of prediction of action boundaries.

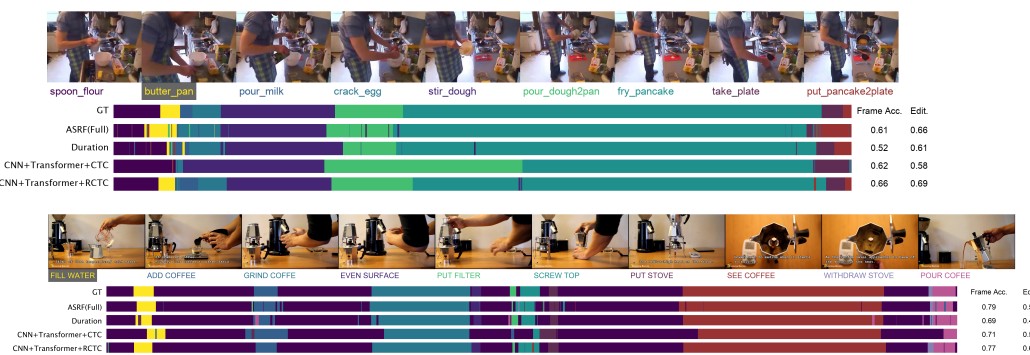

**Figure 8.** Frame and alignment accuracy on randomly selected testing videos from Breakfast (**Top**) and YouTube Instructions (**Bottom**), respectively. (Note: there is a background class (▬) in YouTube Instructions, usually present in between neighboring steps, whose fraction varied from 46% to 83% across different tasks, whereas Breakfast is tightly trimmed).

## 6. Conclusions

We present Hybrid Reduced-CTC to align actions in instructional videos. The main contribution is a hybrid convolutional transformer LSTM to capture long-term temporal dynamics within and between actions, as well as the modified dynamic programming to recursively sum together every possible alignment and reduce the decoding search space by weighting the priority of qualified paths by their visual consistency with existing labels. Our results were found to be competitive in terms of accuracy on four publicly available datasets of different size and difficulty. Based on the confirmed effectiveness of CTC in frame- and transcript-level localization, we plan to further explore its impact on the stability of convergence, due to the observation that that CTC empirically tends to become more difficult to converge than non-CTC models as the length of the input video increases, which suggests more effective under-sampling techniques may be needed to adapt CTC with longer time span.

**Author Contributions:** Conceptualization, L.W. and X.W.; methodology, L.W.; software, L.W.; validation, L.W., A.H., and X.W.; formal analysis, L.W.; investigation, L.W.; resources, L.W.; data curation, L.W.; writing—original draft preparation, L.W.; writing—review and editing, A.H.; visualization, L.W.; supervision, A.H.; project administration, A.H.; funding acquisition, X.W., Y.X., and A.H. All authors have read and agreed to the published version of the manuscript.

**Funding:** This research was funded by Fundamental Research Funds for the Central Universities grant numbers WK2150110007 and WK2150110012, and the National Natural Science Foundation of China grant numbers 61772490, 61472382, 61472381, and 61572454.

**Institutional Review Board Statement:** Not applicable.

**Informed Consent Statement:** Not applicable.

**Data Availability Statement:** YouTube Instructional is publicly available at https://www.di.ens.fr/willow/research/instructionvideos, YouCook2 is publicly available at http://youcook2.eecs.umich.edu, Breakfast is publicly available at https://serre-lab.clps.brown.edu/resource/breakfast-actions-dataset, and 50 Salads is publicly available at http://cvip.computing.dundee.ac.uk/datasets/foodpreparation/50salads.

**Acknowledgments:** We would like to thank the unknown reviewers for their effort and time spent on this work. Hardware platform (a workstation with GeForce GTX Titan Z GPU and 6 GB RAM) and software distribution license (Matlab Deep Learning Toolbox) were provided by the Super Computing Center and Network Information Center of University of Science and Technology of China, respectively. The 50 Salads dataset is distributed by the Computer Vision Image Processing (CVIP) group at the University of Dundee, Scotland, UK; YouTube Instructions dataset is distributed by the Willow project team at the Computer Science Department of the Ecole Normale Superieure (DI ENS), Paris, France. The Breakfast Actions Dataset is distributed by the Serre Lab at Brown University. YouCook2 is distributed by University of Michigan.

**Conflicts of Interest:** The authors declare no conflicts of interest. The funders had no role in the design of the study; in the collection, analysis, or interpretation of data; in the writing of the manuscript; or in the decision to publish the results.

## Abbreviations

Abbreviations
The following abbreviations are used in this manuscript:

| | |
|---|---|
| CTC | Connectionist Temporal Classification |
| LSTM | Long Short-Term Memory Network |
| CR-CTC | Reduced Connectionist Temporal Classification |
| FC Fully | Convolution |
| TM | Transformer |
| ED-TCN | Encoder–Decoder Temporal Convolutional Network |

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
