# Peer review of "End to End Alignment Learning of Instructional Videos with Spatiotemporal Hybrid Encoding and Decoding Space Reduction"

_applsci, doi:10.3390/app11114954_

Round 1
Reviewer 1 Report
No further comments.
Reviewer 2 Report
Please see the attached PDF file.

Reviewer 3 Report
The paper describes a new method for actions alingnment in instructional videos. To this end, they proposed the use of a hybrid convolutional bi-directional LSTM.
The paper in general is well written and well structured. The introduction present a well known concepts but there is a lack of references.
The related work is well explained and the background can be clearly obtained.
The points 3, 4 and 5 are rigorously presented from the mathematical point of view, and experiment section is well raised. Maybe, the point 3 and 4 could be merged in a unique section.
Finally, the conclusions are correct and they resume well their contributions
Round 2
Reviewer 2 Report
The experimental results are updated. From an application viewpoint, the paper stands good.
This manuscript is a resubmission of an earlier submission. The following is a list of the peer review reports and author responses from that submission.
Round 1
Reviewer 1 Report
The manuscript describes a novel method that is able to provide frame-accurate activity boundaries from video, even if not provided with fully annotated training data.
The method seems reasonable, overall I think the research behind the manuscript is pretty decent. However, I am bothered with few things that I wish that authors fix before the paper is accepted for publication.
- Experimental design is not well described, especially because the authors compare their method with methods that work slightly differently (e.g. their method is competitive, but relies on less accurate training data). Please clearly specify what are the input data for training and testing of your method. Is it:
a) Training needs sequence of actions, and a video, no per frame annotations, testing generates per frame results from video only.
b) Training needs sequence of actions, and a video, no per frame annotations, testing generates per frame results from video AND sequence of actions.
I assume it is a). Please clarify that. - Related work section is a bit short. I see you provided a minimum of relatively novel work, but in light of previous comment (1), I would like to have better overview of what method needs what data. Please make the related work section bit longer, include more references, but most importantly please clearly specify the type of input and output data that each competing method needs or provides, as this is the main advantage of your method.
- Presentation of the results - as I understand, your method does not outperform others, but needs far less accurate input data. Currently the results are not well commented, please describe the results more clearly. I hope my understanding is correct, but the result really need more clear commentary.
Minor things:
4. Please make another pass regarding the language. The manuscript is otherwise ok, but right at the beginning, in lines 2 and 3 you have "in order to save the consuming efforts to accurately annotate the temporal
3 boundaries of each action.". What are consuming efforts? Time-consuming perhaps?
5. Under the figure 2, there is a text that says "See Section 5.0.0.3", this is clearly broken reference with two zeroes?
Reviewer 2 Report
Please see the attached pdf files for the comments.

Reviewer 3 Report
The authors present new method for actions alignment in instructional videos using bi-directional LSTM in order to capture long-term temporal dynamics of the actions.
In general, the paper is well written and structured.
The evaluation methodology and mathematical background are rigorously presented.
The related work and result analysis sections could be extended.
Summarizing, it is a good paper with a valuable contribution.
